# Acute Coronary Syndromes and Covid-19: Exploring the Uncertainties

**DOI:** 10.3390/jcm9061683

**Published:** 2020-06-02

**Authors:** Marco Schiavone, Cecilia Gobbi, Giuseppe Biondi-Zoccai, Fabrizio D’Ascenzo, Alberto Palazzuoli, Alessio Gasperetti, Gianfranco Mitacchione, Maurizio Viecca, Massimo Galli, Francesco Fedele, Massimo Mancone, Giovanni Battista Forleo

**Affiliations:** 1Department of Cardiology, ASST-Fatebenefratelli Sacco, Luigi Sacco Hospital, 20157 Milan, Italy; marco.schiavone@unimi.it (M.S.); alessio.gasperetti93@gmail.com (A.G.); mitacchione.gianfra@asst-fbf-sacco.it (G.M.); viecca.maurizio@asst-fbf-sacco.it (M.V.); giovanni.forleo@unimi.it (G.B.F.); 2University of Milan, 20122 Milan, Italy; cecilia.gobbi@unimi.it; 3Department of Medical-Surgical Sciences and Biotechnologies, Sapienza University of Rome, 04100 Latina, Italy; giuseppe.biondizoccai@uniroma1.it; 4Mediterranea Cardiocentro, 80122 Naples, Italy; 5Department of Medical Sciences, Division of Cardiology, AOU Città della Salute e della Scienza, University of Turin, 10126 Turin, Italy; fabrizio.dascenzo@gmail.com; 6Cardiovascular Diseases Unit, Department of Medical Sciences, AOUS Le Scotte Hospital, University of Siena, 53100 Siena, Italy; palazzuoli2@unisi.it; 7Department of Infectious Diseases, ASST-Fatebenefratelli Sacco, Luigi Sacco Hospital, 20157 Milan, Italy; massimo.galli@unimi.it; 8Luigi Sacco Department of Biomedical and Clinical Sciences, University of Milan, 20157 Milan, Italy; 9Department of Clinical Internal, Anesthesiological and Cardiovascular Science, Sapienza University of Rome, 00161 Rome, Italy; francesco.fedele@uniroma1.it

**Keywords:** acute coronary syndromes, myocardial infarction, STEMI, Covid-19, infectious disease, respiratory infections, pathophysiology, percutaneous coronary intervention, thrombolysis, drug treatment

## Abstract

Since an association between myocardial infarction (MI) and respiratory infections has been described for influenza viruses and other respiratory viral agents, understanding possible physiopathological links between severe acute respiratory syndrome coronavirus 2 (SARS-CoV-2) and acute coronary syndromes (ACS) is of the greatest importance. The initial data suggest an underestimation of ACS cases all over the world, but acute MI still represents a major cause of morbidity and mortality worldwide and should not be overshadowed during the coronavirus disease (Covid-19) pandemic. No common consensus regarding the most adequate healthcare management policy for ACS is currently available. Indeed, important differences have been reported between the measures employed to treat ACS in China during the first disease outbreak and what currently represents clinical practice across Europe and the USA. This review aims to discuss the pathophysiological links between MI, respiratory infections, and Covid-19; epidemiological data related to ACS at the time of the Covid-19 pandemic; and learnings that have emerged so far from several catheterization labs and coronary care units all over the world, in order to shed some light on the current strategies for optimal management of ACS patients with confirmed or suspected SARS-CoV-2 infection.

## 1. Introduction

In December 2019, an outbreak of pneumonia caused by a novel coronavirus occurred in Wuhan, Hubei province, spreading rapidly first throughout China, and subsequently across Europe, the United States (US), and the rest of the world [1,2,3], reaching a total number of 3,435,894 confirmed cases worldwide as of 5 May 2020 [4]. On 30 January 2020, the World Health Organization (WHO) declared the Covid-19 outbreak a public health emergency of international concern, and on 12 March -2020, it was characterized as a pandemic. Patients exposed to this virus, named severe acute respiratory syndrome coronavirus 2 (SARS-CoV-2), frequently present with fever, cough, and shortness of breath within 2 to 14 days after exposure, and then usually develop coronavirus disease (Covid-19)-related pneumonia [5]. Although respiratory symptoms prevail among all clinical manifestations of Covid-19, preliminary studies showed that some patients may develop severe cardiovascular (CV) damage, while other patients with underlying CV diseases might have an increased risk of death [5,6,7]. 

Moreover, the Covid-19 outbreak has put a lot of pressure on overloaded healthcare systems, especially in Lombardy (Italy) and more generally in Northern Italy, where Covid-19 has spread very rapidly, causing concern regarding the capacity of the system to respond to the needs of intensive care treatments [8]. All possible efforts have been made in order to give the maximum number of patients the chance to be admitted and treated in hospitals. All non-urgent procedures have been cancelled and routine clinical practice has been completely modified. In the context of an overwhelmed healthcare system, screening and elective treatments of coronary artery disease (CAD) have been underestimated, meaning dealing with acute coronary syndromes (ACS) has become more complicated and apparently less frequent. Nevertheless, ACS still remain a major cause of morbidity and mortality worldwide and are responsible for more than 1 million hospital admissions in the US annually, while ischemic heart disease accounts for almost 1.8 million annual deaths, or 20% of all deaths in Europe, although with large variations between different European countries [9,10].

During this pandemic, links between SARS-CoV-2 and ACS have not been established yet and a common guidance on how to handle ACS in Covid-19 and non-Covid-19 patients is needed. The aim of this review is to shed some light on those issues, analyzing possible physiopathological links between Covid-19 and ACS and evaluating the best strategy to balance optimal ACS management and infectious risks related to Covid-19 outbreak.

## 2. Acute Coronary Syndromes and COVID-19

### 2.1. Pathophysiology 

Acute coronary syndromes (ACS) reflect a spectrum of pathological conditions compatible with acute myocardial ischemia or infarction that are usually due to an abrupt reduction in coronary blood flow [11]. The clinical spectrum of ACS may range from myocardial infarction with ST-segment elevation (STEMI), which generally reflects an acute total coronary occlusion, to myocardial infarction without ST-segment elevation (NSTEMI) or unstable angina (UA), with or without myocardial injury, respectively [12]. The current fourth universal definition defines myocardial infarction (MI) as the presence of acute myocardial injury, detected by an elevated cardiac troponin (cTn) value above the 99th percentile of the upper reference limit (URL), in the setting of evidence of acute myocardial ischemia related to symptoms, electrocardiogram (ECG), imaging changes, or angiographic findings [13]. Furthermore, cTn I and T, regulatory components of the contractile apparatus of myocardial cells, are the preferred biomarkers for the evaluation of myocardial injury and have been used worldwide. It should be underlined that any type of myocardial injury can result in significant cTn release into the blood, but cTn elevation does not allow for discrimination between the underlying pathophysiological mechanisms [14]. Several clinical conditions related to the mismatch between oxygen supply and demand, such as respiratory failure (predominantly hypoxaemia) and infectious disease (particularly sepsis), may induce or lead to myocardial injury or to type 2 MI [13,15,16]. Mechanisms related to myocardial injury are summarized in Table 1. 

Identification of type 2 MI can be challenging due to more frequent atypical clinical presentations (such as with dyspnoea), higher prevalence of comorbidities that may mask ischemia [17], and lower frequency of ischemic electrocardiographic findings (Q waves or new ST-T wave changes) and new regional wall motion abnormalities. Moreover, culprit lesions can be identified in a small percentage of cases by coronary angiography [18,19,20,21]. Nowadays, it is well accepted that sepsis and other infections are associated with CV events, especially ACS [22,23]. In particular, the risk of MI in the context of respiratory infectious disease reaches a peak at the onset of infections and is proportional to the severity of illness [22]. Acute respiratory failure with consequent severe hypoxaemia contributes to reduce oxygen supply and determines activation of the sympathetic system, which increases heart rate, cardiac output, and contractility, factors that can increase myocardial oxygen demand [24,25]. Incidence of myocardial injury or infarction in critically ill patients may go unrecognized [26], as post-mortem studies have suggested, where a prevalence of undiagnosed acute myocardial infarction (AMI) ranging from 5% to 25% in patients who died from acute respiratory failure was observed [27,28]. 

Another possible mechanism implicated in the association between respiratory tract infections and ACS is the pro-inflammatory state. Since this association has been established for a variety of pathogens and sites of infection, it is likely that the causal agent and the host response could have a crucial role in eliciting an inflammatory pattern that may trigger ACS [22]. Atherosclerotic plaques contain inflammatory cells that proliferate and secrete cytokines that stimulate smooth muscle cells to form a fibrous cap [29]. An inflammatory state at any other site generates circulating cytokines, such as interleukins 1, 6, and 8 and tumor necrosis factor α, which can activate inflammatory cells in atherosclerotic plaques [30]. Studies in murine models [31] and post-mortem studies in humans [32] have shown that inflammatory activity in atheromatous plaques increases after an infectious stimulus. When activated, intraplaque inflammatory cells, especially macrophages and T-cells, upregulate host response proteins, including metalloproteinases and peptidases, which degrade components of the extracellular matrix and promote an oxidative burst, all of which contribute to destabilization of plaques [33,34]. When the plaque surface becomes disrupted, thrombogenic elements (collagen, phospholipids, tissue factor, and platelet-adhesive matrix molecules) are exposed, leading to the acute formation of a thrombus, which is the typical mechanism related to type 1 MI [35]. Moreover, inflammation promotes a prothrombotic state, which could further increase the risk of microangiopathy in multiple organs [36] and coronary thrombosis at sites of plaque disruption [37]. The inflammatory reaction in the coronary arteries impairs fibrinolysis through the inhibition of action of antithrombin, protein C system, and tissue factor pathway inhibitor, three major coagulation-inhibiting proteins that facilitate thrombosis [38,39]. Finally, influenza viruses and other viral respiratory infections are associated with expression of genes that have been linked to platelet activation and to an increased risk of MI [40].

### 2.2. ACS and Other Acute Respiratory Infections 

In the early 20th century, an excess mortality during influenza and pneumonia epidemics was first recognized [41], but the specific association with influenza or other respiratory infections and MI was not described until decades later (see Table 2). 

More recent studies have documented an increased risk of MI with influenza, pneumonia, acute bronchitis, and other chest infections [42,43,44]. In retrospective and prospective case series, a rate of CV events of about 30% and a rate of MI of about 8% were found among patients who were hospitalized for community-acquired pneumonia [45,46]. Other retrospective studies suggested that hospitalization for pneumonia was associated with both short and long-term increased risk of CV events; in an analysis performed by Medina et al., 318 out of 1271 patients (25%) developed CV events over 10 years after pneumonia hospitalization [47]. A meta-analysis of 10 case–control studies conducted by Barnes et al. demonstrated a two-fold increased risk of AMI in patients with recent influenza infection or respiratory tract infection; a recent influenza infection, influenza-like illness, or other respiratory tract infection was significantly more likely in AMI cases, with a pooled odds ratio (OR) of 2.01 (95% confidence interval CI: 1.47 to 2.76) [53]. From a large American database, among 1, 884, 985 admissions for AMI from January 2013 to December 2014, influenza or other viral respiratory infections accounted for 1.1% of the patients (9885 and 11485 patients respectively) and were associated with worse outcomes and higher in-hospital case fatality (approximately 13%) [48]. 

Vejpongsa et al. also showed that AMI patients with concomitant influenza infection or other viral respiratory infections were less likely to receive cardiac catheterization across all age groups when compared with patients with AMI alone (22% vs. 43.8% vs. 58.8%, *p* < 0.001) [48]. However, amongst those patients who underwent coronary catheterization in the three different groups, more than half required revascularization; the rate of revascularization was lower in those with concomitant influenza than those without, suggesting that clinicians had appropriately identified which patients should undergo coronary angiography [48]. These interesting findings should be highlighted and related to what is currently happening during this Covid-19 pandemic, given that patients infected with SARS-CoV-2 virus seem to undergo cardiac catheterization less, likely due to high risk of infection spreading.

Two other studies have evaluated the prevalence of respiratory infections and influenza among patients with angiographically confirmed MI [49] and STEMI [50], respectively. Ruane et al. confirmed that respiratory infections can trigger MI [50] and Caussin et al. showed that influenza epidemic may be associated with a significant excess relative risk of STEMI [49]. These results have to be underlined, since these authors only analyzed cases with angiographically confirmed MI, limiting the potential bias of other studies based on retrospective database analysis related to the inclusion of cases with myocardial injury (potentially misdiagnosed as MI), especially considering the association between cTn elevation and sepsis.

Acute coronary syndromes and MI were also noted to occur in severe acute respiratory syndrome (SARS), an infectious disease that afflicted a total of 8096 people in 29 countries in 2003, with a mortality around 9.6% [54]. In a prospective study of 75 patients hospitalized with SARS, AMI was the cause of death in 2 out of 5 fatal cases [51]. A study from Singapore reported post-mortem examinations in 8 patients who died suddenly and unexpectedly from SARS; 1 out of 8 patient had subendocardial infarction with occlusive coronary disease (as well as AMI on presentation with SARS), while 4 patients had pulmonary thromboembolism and 1 patient developed marantic valvular vegetations, along with infarction in the heart, kidneys, spleen, and brain [52]. These findings suggest a possible link between severe acute respiratory syndromes, thrombophilia, and subsequently ACS. Additionally, Middle East respiratory syndrome (MERS), which was first reported in 2012 in Saudi Arabia and has afflicted 2519 patients with 866 associated deaths (case-fatality rate: 34.4%) in 27 countries [55], has been related to CV diseases. A systematic analysis of 637 MERS patients showed that 30% of cases had underlying cardiac diseases, 50% had hypertension, 50% had diabetes, and 16% had obesity [56]. The clinical risk factors for mortality in MERS were older age, male sex, and CV-related underlying medical conditions, including hypertension, diabetes, cardiac diseases, and chronic kidney disease [57,58,59]. Data on the incidence of ACS in the context of MERS infection are lacking. Alhogbani reported a case of a 60-year-old patient with MERS coronavirus (MERS-CoV) infection who presented with respiratory symptoms, chest pain, TnT elevation, diffuse T-wave inversion, and severe left ventricular (LV) dysfunction; acute myocarditis was then diagnosed with cardiac magnetic resonance, which excluded an ischemic cardiomyopathy [60]. 

A different relative risk of MI with several respiratory infections has been described by Kwong et al. [43]. Incidence ratios for AMI within 7 days after detection of influenza B, influenza A, respiratory syncytial virus, and other viruses were 10.11 (95% CI, 4.37 to 23.38), 5.17 (95% CI, 3.02 to 8.84), 3.51 (95% CI, 1.11 to 11.12), and 2.77 (95% CI, 1.23 to 6.24), respectively. Additionally, Guan et al. analyzed the potential relationship between AMI and previous influenza virus infection, and found that AMI was associated with the presence of IgG antibodies to influenza virus A, B, herpes simplex virus 1 and 2 (HSV1-2), cytomegalovirus (CMV), HSV-1 and HSV-2, adenovirus (ADV), rhinovirus (RV), and chlamydia pneumonia with different OR (in particular, adjusted OR for influenza A 5.5, 95% CI: 1.3–23.0; adjusted OR for influenza B: 20.3, 95% CI: 5.6–40.8) [61]. Those findings suggest a stronger correlation between influenza B virus infection and AMI; however, evidence on the mechanisms that may lead to different AMI rates in relationship with some viral infections are lacking. Limitations regarding laboratory testing, baseline characteristics (comorbidities and vaccinations), and clinical courses of the infections (mild or severe) may affect the estimation of the absolute risk of AMI during or after different viral infections.

Moreover, the pooled results of the aforementioned meta-analysis from Barnes et al. demonstrated an association between influenza vaccination and a lower risk of composite CV events, with a pooled OR of 0.71 (95% CI: 0.56 to 0.91), equating to an estimated vaccine effectiveness of 29% (95% CI: 9% to 44%) against AMI [53]. This finding is in line with other results from another meta-analysis from Udell et al., which showed that the influenza vaccine given to high-risk patients, such as patients with CAD, reduced their risk of a major adverse cardiovascular event (MACE) (patients treated with influenza vaccine and MACE (2.9%) vs. patients treated with placebo or control and MACE (4.7%); RR, 0.64 (95% CI: 0.48–0.86), *p* = 0.003) [62]. Therefore, current European guidelines on the diagnosis and management of chronic coronary syndromes recommend annual influenza vaccination in order to improve prevention of AMI in patients with CAD and decrease CV mortality [63,64,65].

### 2.3. Myocardial Injury and ACS in Patients with Covid-19: What We Know

Although the clinical manifestations of Covid-19 are dominated by respiratory symptoms, evidence of myocardial injury was recognized in early cases in China (see Table 3).

Huang et al. first reported a prevalence of acute myocardial injury of 12% as a major complication among 41 hospitalized patients infected with SARS-CoV-2 [5]. In another study from Wang et al. conducted on 138 hospitalized patients with Covid-19, cardiac injury was found in 7.2% of patients overall and in 22.2% of patients who were treated in the intensive care unit (ICU) [1]. These findings could suggest that acute myocardial injury may have a relevant role in worsening clinical outcomes in Covid-19 patients, even without clear evidence of myocardial ischemia. Indeed, Zhou et al., in a retrospective report of 191 patients admitted with SARS-Cov-2 pneumonia, diagnosed acute myocardial injury in 33 out of 191 (17%) patients in their cohort [66]. Interestingly, they found that non-survivors were more likely to develop acute myocardial injury than survivors (*n* = 32, 59% vs. *n* = 1, 1%; *p* < 0.0001). Notably, the first autopsy in this cohort was performed on a 53-year-old woman with chronic renal failure, and findings were consistent with AMI (data resulting from personal communication between a pathologist and the Chinese Academy of Science, not available in a published manuscript). In a single-center retrospective study by Shi et al., conducted on a cohort of 416 consecutive Covid-19 patients in Wuhan, China, cardiac injury was found in 19.7% patients (*n* = 82) [6]. Those patients were older, had more CV comorbidities (hypertension, diabetes, cerebrovascular disease, and heart failure), and presented with more severe acute illness than patients without cardiac injury. This study demonstrated that myocardial injury was independently associated with an increased risk of mortality in patients with Covid-19. It should be underlined that among those 82 patients with cardiac injury, only 22 (26.8%) underwent an electrocardiogram (ECG) after admission, and only 14 out of 22 ECGs (63.6%) were performed at the same time as the elevation of cardiac biomarkers. All ECGs were described as abnormal, with findings compatible with myocardial ischemia, such as T-wave depression and inversion, ST-segment depression, and Q waves. The above findings may suggest that 14 out of 416 patients in this cohort (3.36%) may have developed myocardial ischemia, with features consistent with NSTEMI. No evidence of STEMI in this cohort was provided, even if limited availability of ECGs may have led to underestimation of AMI cases. In addition, the National Health Commission of China reported that among people who died from Covid-19, 11.8% of patients without underlying CV diseases had substantial heart damage, showing elevated levels of cTnI or cardiac arrest during hospitalization [67]. In a meta-analysis by Lippi et al. that included a total number of 341 patients (123 with severe disease, 36%), it appeared that cTnI values were significantly increased in patients with severe SARS-CoV-2 infection compared to those with milder forms of disease [68].

Several mechanisms that could explain the onset of acute myocardial injury related to myocardial ischemia in SARS-CoV-2 infection have been proposed. Some of them may resemble the ones identified for other respiratory infectious agents, such a pro-inflammatory state and a cytokine storm (which could cause plaque instability), or a prothrombotic state and hypoxaemia-related damage due to acute respiratory failure. The rise in cTn tracks with other inflammatory biomarkers, such as D-dimer, interleukin-6, and lactate dehydrogenase, raises the possibility that this may reflect cytokine storm more than isolated myocardial injury [69]. On the other hand, some reports of patients presenting with cardiac symptoms, ECG changes, or new wall motion abnormalities may suggest a different pattern, such as viral myocarditis and stress cardiomyopathy. The underutilization of coronary angiography during this outbreak due to the high infectious risk makes it more difficult to establish a definite differential diagnosis in many cases.

Furthermore, specific damage caused by SARS-CoV-2 infection might be related to angiotensin-converting enzyme 2 (ACE2) receptors, which have been shown to represent the entry point into human cells for some coronaviruses, such as SARS-CoV and SARS-CoV-2. ACE2 receptors are widely expressed in both the lungs and the CV system; therefore, ACE2- related signalling pathways might also have a role in myocardial injury. At the time of writing, a lot of studies are ongoing all over the world, which will hopefully tell us more about the link between ACE2 receptors, Covid-19, and CV diseases.

## 3. Critical Issues in Management and Treatment of ACS Patients

### 3.1. Where did All the STEMIs Go?

The relevant impact of Covid-19 pandemic is related to the diagnosis and management of patients with ACS who were not hospitalized for confirmed or suspected Covid-19. Diagnosis and treatment of ACS—especially STEMI—start from the point of first medical contact (FMC), defined as the time point when the patient is initially assessed by a physician, paramedic, nurse, or trained medical personnel who can interpret the ECG and deliver medical interventions in the pre-hospital or in-hospital setting [70]. Prompt activation of emergency medical services (EMS) is crucial, since ischemic time duration is a major determinant of infarct size in patients with STEMI, while prompt recognition and early management are critical in reducing morbidity and mortality related to ACS [71]. It has been postulated that in the midst of this healthcare crisis, hospital admissions for ACS have dramatically decreased, mostly due to the fact that patients do not activate EMS because of the “do not come to the hospital” policy and due the fact that hospitals are now perceived as dangerous places. Prof. B. Casadei, Professor of Cardiovascular Medicine at the University of Oxford and European Society of Cardiology (ESC) president, stated that in the worst hit areas, hospital admissions for ACS were reduced by up to 75% [72]. In the US and Spain, approximately 38% and 40% reductions in cardiac catheterization laboratory STEMI activations were experienced [73,74], respectively, while in Italy a reduction in hospitalizations for STEMI (26.5%; less striking than with NSTEMI—65.1%) was reported [75]. Those findings were corroborated by De Filippo et al., who performed a retrospective analysis on consecutive patients who were admitted at 15 hospitals in northern Italy for ACS. They showed that the mean admission rate for ACS during the study period (20 February 2020, to 31 March 2020) was 13.3 admissions per day vs. 18.0 admissions per day during the earlier period in the same year vs. 18.9 admissions per day during the same timeframe of the previous year [76]. 

### 3.2. Are We Really Prioritizing and Treating STEMI Patients the Way We Should?

In Hong Kong, Tam et al. described a small number of patients with STEMI seeking medical help (*n* = 7) and found large delays in presentation after infection control measures were implemented in their country when compared to 2018–2019 presentation times during office and non-office hours, respectively, where median symptom onset to first medical contact = 318 min (IQ range: 75–448) vs. 82.5 min (IQ range: 32.5–195) vs. 91.5 min (IQ range: 32.25–232.75) [77]. The reason proposed for these delays vary and are mostly related to hesitancy to go to the emergency department (ED) or to activate the EMS, introducing a first “Covid-19-related delay” in the so-called “total ischemia time” (Figure 1). 

Tam et al. also postulated that many STEMI patients do not seek care at all, which may contribute to a global underestimation of ACS cases [77]. In Italy—particularly in Lombardy—similarly to in several other European countries and US states, the healthcare system has also been facing a huge overload, causing unbearable consequences on resources for cardiology, since the availability of ward and cardiology care unit (CCU) beds have been drastically reduced and admissions for elective procedures have been suspended (such as coronary catheterizations). The EMS is now focused on dealing with the Covid-19 outbreak, so less resources are available to cope with other emergencies such as ACS, due to limited means of support and healthcare staff, who are either sick or committed to handling the Covid-19 emergency. In order to avoid SARS-CoV-2 spreading, on 8 March 2020, the government of Lombardy identified 13 hospitals with catheterization laboratories acting as *“hubs”*, with the remaining hospitals acting as *“spokes”*, in order to gather cardiovascular emergencies in certain coronary care units (CCUs) all across the region [78]. About 10 million people live in Lombardy, representing one-sixth of the Italian population, meaning that this reorganization introduced another “Covid-19-related delay” (Figure 1) in managing patients with STEMI, who were potentially more distant from a *“hub”* catheterization laboratory when activating EMS. Moreover, *“hubs”* must have more than 1 catheterization lab, and at least 1 of those should be dedicated to suspected or diagnosed Covid-19 patients, so that the most appropriate protocol can be followed. Major impacts of this healthcare policy are expected, since delays in seeking and delivering care due to patient fears of contracting an infection from the healthcare system and longer times taken to reach *“hubs”* could have harmful impacts on ACS patients outcomes.

During this pandemic, finding a balance between risks related to untimely treatment of ACS patients and SARS-CoV-2 infection control has become a global challenge. Commonly, regional reperfusion strategies are established to maximize efficiency in treatments, since primary percutaneous coronary intervention (PCI)—bypassing the ED—is the routine treatment for STEMI patients [79]. Several trials and meta-analyses endorsed by European and American Guidelines have clearly established the superiority of primary PCI compared to thrombolysis over the years. As early as 1997, a quantitative review published by Weaver et al. based on outcomes at hospital discharge or at 30 days demonstrated that primary PCI was superior to thrombolytic therapy for treatment of patients with AMI (*n* = 2606); mortality at 30 days or less was 4.4% for patients treated with primary PCI (*n* = 1290) compared with 6.5% for patients treated with thrombolysis (*n* = 1316), representing a 34% reduction (OR, 0.66; 95% CI, 0.46–0.94; *p* = 0.02) [80]. More recently, a pooled analysis of 22 randomized trials (total patients *n* = 6763) by Boersma et al. demonstrated that primary PCI was associated with significantly lower 30-day mortality rate compared to thrombolysis (adjusted OR, 0.63; 95% CI (0.42–0.84)), regardless of treatment delay [81].

Despite this clinical evidence, to cope with the abrupt Covid-19 outbreak, case decisions were individualized at the beginning of the pandemic, taking into account the risk of SARS-CoV-2 exposure versus the risk of delaying diagnosis or therapy. Subsequently, Peking Union Medical College Hospital and Sichuan Provincial People’s Hospital proposed recommendations in China, which are summarized as follows [82,83]: with regard to STEMI patients, thrombolytic therapy was recommended over primary PCI if Covid-19 was confirmed or could not be excluded within a short time, while for NSTEMI–UA, the priority was to exclude SARS-CoV-2 infection first, since door-to-balloon time is less crucial in those patients than in STEMIs. These recommendations were endorsed by Daniels et al., who stated that thrombolysis might be the best compromise of prompt reperfusion for the patient, buying time for a complete diagnosis to be made [84]. According to Peking’s protocol, blood tests, pharyngeal swab or sputum specimen, or blood samples should be performed for detection of SARS-CoV-2 nucleic acid before starting treatment, in addition to chest computed tomography (CT) scanning and evaluation by infectious disease specialists [82]; while Sichuan’s protocol relies on rapid nucleic acid testing before starting care. These recommendations are undoubtedly useful for minimizing and controlling the spread of SARS-CoV-2 infection, but data on the outcomes of ACS patients are needed to confirm that delaying treatment and use of thrombolysis as a first therapy to treat STEMI in confirmed or suspected Covid-19 patients are not associated with worse outcomes. 

Indeed, despite Lombardy being considered an area with cluster transmission of SARS-CoV-2 since late February 2020, most *“hubs”* are performing primary PCI without waiting for screening test results in order to avoid important delays and reliance on thrombolysis. Stefanini et al. performed a retrospective analysis on 28 Covid-19 patients who were admitted for STEMI: they found that 24 patients (85.7%) did not have a SARS-CoV-2 test result at the time of coronary angiography and that 11 patients (39.3%) did not have obstructive CAD [85]. In line with the aforementioned analysis, in our tertiary care center (“Ospedale Luigi Sacco”, Milan, Italy) located in the heart of the Italian epidemic, no cases of ACS requiring PCI were reported among more than 900 patients admitted for SARS-CoV-2 infection, as of May 5, 2020, suggesting a possible link between SARS-CoV-2 and type 2 MI or myocarditis–stress cardiomyopathy. A case report from Hu et al. described a 37-years-old male patient presenting with chest pain and dyspnea, with ST-segment elevation in the inferior leads, elevation of TnT, and severe depression of LV ejection fraction (27%), in which an emergency coronary computed tomography angiography (CCTA) revealed no coronary stenosis and a diagnosis of fulminant myocarditis was made [86]. Another case report from Inciardi et al. described a patient infected with SARS-CoV-2 that had severe fatigue, no chest pain, minimal diffuse ST-segment elevation (more prominent in the inferior and lateral leads), an ST-segment depression with T-wave inversion in lead V1 and aVR, severe LV dysfunction, and no evidence of obstructive CAD at time of urgent coronary angiography, who was then diagnosed with acute myopericarditis [87]. A first case series from New York City (USA) described 18 Covid-19 patients with ST-segment elevation indicating potential AMI; among those patients, 9 (50%) underwent coronary angiography, 6 out of 9 (67%) had obstructive disease, and 5 (56%) underwent PCI [88]. All these findings should be underlined by considering that in such cases, thrombolytic therapy—if used—may have increased the hemorrhagic risk without adding any benefit on the ischemic side. Since reperfusion seems not to be mandatory in a great number of patients (possibly due to the previously highlighted link between respiratory infections and type 2 MI), relying on systematic thrombolysis seem not to be justified from these initial European and American reports [85,88]. Hence, also based on those findings, the Society for Cardiovascular Angiography and Interventions (SCAI), American College of Cardiology (ACC), American College of Emergency Physicians (ACEP), and American Heart Association (AHA) published guidance on the management of AMI during the Covid-19 pandemic in the US [89,90]. These guidelines state that after a first evaluation in the ED to assess the infectious risks, STEMI patients should undergo primary PCI whenever possible if it can be provided within an adequate time frame from the symptoms onset and STEMI diagnosis. STEMI patients should be transferred to the catheterization laboratory as rapidly as possible, and although door-to-balloon times are expected to be longer during the Covid-19 pandemic, a primary PCI strategy should remain the first choice. Thrombolytic therapy should not be the standard of care strategy and should be limited to particular situations, such as in non-PCI capable hospital or when PCI cannot be performed within an acceptable time frame. Those latest recommendations are more consistent with general European and American Guidelines on STEMI [70,91], confirming that primary PCI remains the reperfusion therapy of choice if feasible within an acceptable time frame from STEMI diagnosis. 

In summary, protocols should guarantee the feasibility of performing PCI in facilities approved for treatment of Covid-19 patients, avoiding potentially harmful thrombolysis, in compliance with adequate safety measures to protect healthcare workers (see after), since primary PCI should remain the default strategy in patients with clear evidence of a STEMI, as clearly stated in American guidelines on the management of AMI during the Covid-19 pandemic [89]. An efficacious strategy could be to organize separated catheterization labs and subsequently CCUs or cardiology wards for patients with and without SARS-CoV-2 infections, although this may be possible only in high volume hospitals. On 3 April 2020, SCAI and the Canadian Association of Interventional Cardiology (CAIC) announced the formation of the North American Covid-19 ST-Segment Elevation Myocardial Infarction Registry (NACMI) [92], which hopefully will tell us more about this topic, since further data are needed to detect and characterize patients with STEMI and Covid-19 and to optimize treatment.

### 3.3. Organization Issues for Workers and Catheterization Labs

Catheterization lab staff needs time to set up protective gear. According to latest recommendations, appropriate personal protective equipment (PPE) should include gowns, surgical gloves, protective eyewear, full face shields, disposable caps, shoe covers, and a N95/99/100 mask [93,94,95]. However, this perspective has mostly formed by the experience with SARS in 2005. Although protection of healthcare workers is essential, especially during this outbreak, which is seeing high rates of infections among healthcare personnel [96], this setting-up may contribute to introducing another “Covid-19 related delay” in treating STEMI (Figure 1). Tam et al., in a previously mentioned letter, found that device times in catheterization labs were higher during the Covid-19 outbreak when compared to 2018–2019 times during office and non-office hours, respectively (catheterization lab: 33 min (IQ: range 21–37) vs. 20.5 min (IQ: range 16–27.75) vs. 24 min (IQ: range 18–30), respectively) [77]. Importantly, most catheterization labs have either normal or positive ventilation systems and are not designed for infection isolation, meaning that terminals must be cleaned following the procedure is needed, leading to eventual further delays for subsequent procedures [93]. If possible, in order to avoid SARS-CoV-2 spreading, critical patients should be intubated if needed prior to arrival at the catheterization laboratory.

### 3.4. Drug Treatment 

Among drugs commonly used to treat ACS patients, care should be taken when administering antiplatelet therapy. Clopidogrel and ticagrelor have specific interactions with lopinavir–ritonavir, a combination of antiviral drugs that have previously been used to treat SARS and MERS, having inhibitory activity against SARS-CoV and MERS-CoV in vitro and in an animal models [97,98]. In though in a randomized, controlled, open-label trial conducted by Cao et al., hospitalized adult patients with severe Covid-19 showed no benefit with lopinavir–ritonavir treatment beyond standard care, this drug combination is still used worldwide and is awaiting future trials that may help to confirm or exclude the possibility of a treatment benefit [99]. Lopinavir–ritonavir should not be used in combination with clopidogrel or ticagrelor due to their potent CYP3A4 inhibition [100], which results in a diminished effect of clopidogrel and an increased effect of ticagrelor; prasugrel should be used if no contraindications are present, or a testing-guided approach to evaluate platelet function may be considered [101]. Despite some concerns that were raised at the beginning of the pandemic [102], no evidence of severe adverse events, long-term survival, acute health care utilization, or quality of life in patients with Covid-19 using aspirin and other non-steroidal anti-inflammatory drugs (NSAIDS) has been provided, as stated by the WHO [103]. It may be assumed that low-dose aspirin can be safely used as antiplatelet drug in Covid-19 patients.

Additionally, atorvastatin and rosuvastatin should be started at the lowest possible dose when coadministered with lopinavir–ritonavir, since these antiviral drugs inhibit CYP3A4, OATTP1B1, and BCRP, which have roles in the metabolism of statins [101]. Beta-blockers—especially metoprolol—should be administered cautiously in patients consuming chloroquine or hydroxychloroquine due to CYP2D6 inhibition [104] and the potential role of hydroxychloroquine in reducing heart rate [105]. 

Renin–angiotensin–aldosterone system (RAAS)-related drugs (such as ACE inhibitors and angiotensin II receptor blockers, ARBs) are a cornerstone of therapy after MI, since maintenance of therapy in the days to weeks after the index event has been shown to reduce early mortality [106]. Despite a lack of evidence of drug consumption or discontinuation in these patients included in the previous mentioned studies, it has been hypothesized that abrupt withdrawal of RAAS inhibitors in high-risk patients, especially those who have heart failure or previous MI, may result in clinical instability and adverse outcomes [107,108] and may eventually be related to myocardial injury.

Antivirals drugs used for SARS-CoV-2 infection treatment may have potential interactions with oral anticoagulants (OACs). Several case reports have highlighted the necessity of augmented doses of warfarin in patients treated with ribavirin; the international normalized ratio (INR) should be monitored carefully in these patients [109,110]. Due to the inhibitory effect of lopinavir–ritonavir on CYP3A4, which is involved in the hepatic clearance of some novel OACs, rivaroxaban should be avoided and apixaban should be administered at 50% of the standard dose [101]. Given the potential interactions, low molecular weight heparins or unfractionated heparin should be preferred over OACs; moreover, the first evidence showed decreased mortality in most severe Covid-19 patients with coagulopathy [111]. Drugs that are potentially useful for treatment of acute coronary syndromes and coronary artery disease and their potential interactions are summarized in Table 4.

## 4. Conclusions

Despite being eclipsed by Covid-19 outbreak, acute coronary syndromes are still a major cause of morbidity and mortality worldwide and should not be overshadowed in this era, especially because of the possible physiopathological links (currently unexplored) with SARS-CoV-2 infection.

Given the limited heterogeneity of data published in recent months, the potential overlapping symptomatology between ACS and SARS-CoV-2 infection, and the underestimation of ACS cases during Covid-19 outbreak, more reliable data are needed to estimate the real prevalence of ACS during this pandemic. Although reports to date suggest that cTn elevation in Covid-19 may be related more to myocardial injury or type 2 MI than to type 1 MI, consistent with our analysis, more data are needed to properly understand all mechanisms that may induce ACS in SARS-CoV-2 infection.

All efforts made during the last decades to develop strategies to facilitate transfer of ACS patients in whom AMI is suspected directly to the hospital offering 24/7 PCI-mediated reperfusion therapy should not be forgotten. Specific protocols to balance infective risks related to Covid-19 and optimal ACS management should be implemented (especially STEMI), without delays and with preferential PCI treatment whenever possible.

## Authors Contributions

M.S., C.G., M.M., and G.B.F. conceived of this review. M.S., C.G., and A.G. structured and organized this review. M.S. and C.G. revised the literature and synthesized study data. M.S. and C.G. wrote the original draft of this paper. M.M. and G.B.F. supervised the entire work as senior authors. G.B.-Z., F.D., A.P., M.V., F.F., and M.M. provided expert commentary on how to manage ischemic patients during the Covid-19 pandemic due to their expertise in treating coronary artery disease and myocardial infarction. M.G. provided expert commentary on links between respiratory infections and myocardial involvement, and on infectious risk in ischemic patients during Covid-19 pandemic due to his expertise in infectious disease. G.B.-Z., F.D., A.P., A.G., G.M., M.V., M.G., F.F., M.M., and G.B.F. revised and edited the original draft of this paper. M.S. and C.G. updated this review by analyzing the latest published studies and reports. All authors have read and approved the submitted version. 

## Figures and Tables

**Figure 1 jcm-09-01683-f001:**
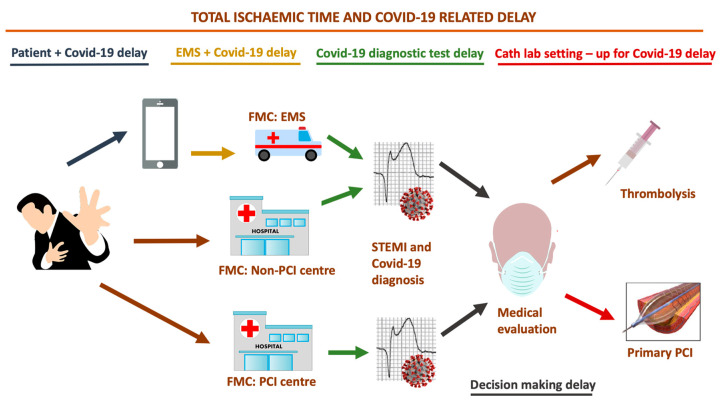
“Covid-19 related delays” in treating ST-elevation myocardial infarction (STEMI) patients. Adapted from Ibanez et al., 2017 European Society of Cardiology Guidelines for the management of acute myocardial infarction in patients presenting with ST-segment elevation [70]. **Abbreviations:** Covid-19: coronavirus disease; EMS: emergency medical services; FMC: first medical contact; PCI: percutaneous coronary intervention; STEMI: ST-elevation myocardial infarction.

**Table 1 jcm-09-01683-t001:** Mechanisms of myocardial injury. Adapted from Thygesen et al. Fourth universal definition of myocardial infarction (2018) [13].

Myocardial Injury		
Related to Primary Acute Myocardial Ischemia	Related to Oxygen Supply/Demand Imbalance	Other Causes
Plaque rupture—erosion with occlusive thrombosis Plaque rupture—erosion with non-occlusive thrombosis	**Reduced myocardial perfusion**	**Cardiac conditions**
	Coronary artery spasm Microvascular dysfunction Coronary embolism Coronary artery dissection Sustained bradyarrhythmia Hypotension or shock Respiratory failure with hypoxaemia Severe anaemia	Heart failure Myocarditis Cardiomyopathy (any type) Takotsubo syndrome Coronary revascularization procedure Cardiac procedure other than revascularization Catheter ablation Defibrillator shocks Cardiac contusion
	**Increased myocardial oxygen demand**	**Systemic conditions**
	Sustained tachyarrhythmia Severe hypertension with or without left ventricular hypertrophy	Sepsis, infectious disease Chronic kidney disease Stroke, subarachnoid hemorrhage Pulmonary embolism, pulmonary hypertension Infiltrative diseases, e.g., amyloidosis, sarcoidosis Chemotherapeutic agents Critically ill patients Strenuous exercise

**Table 2 jcm-09-01683-t002:** Acute coronary syndromes and other acute respiratory infection—main studies.

Study, Year, Journal	Infection	Population and Timeline	Myocardial Infarction Diagnosis	Cases with Myocardial Infarction and Respiratory Infectious Disease
**Smeeth et al., 2004, New England Journal of Medicine** [42]	Systemic respiratory tract infection (pneumonia, acute bronchitis, chest infections, and influenza)	MI diagnosed 91 days after infection exposure	UK GPRD data (1495 cases were excluded because the date of the MI was uncertain)	MI: *n* = 3254 Respiratory infectious disease: *n* = 20,921
**Kwong et al., 2018, New England Journal of Medicine** [43]	Influenza A/B, RSV, adenovirus, CoV, enterovirus (including rhinovirus), HPIV, and HMPV	Admission for MI within 7 days after laboratory confirmation of influenza	ICD-10 diagnostic code	MI: *n* = 364 Respiratory infectious disease: *n* = 19,045
**Warren-Gash et al., 2013, British Medical Journal** [44]	Influenza A H1N1	Respiratory tract infection developed within one month before admission for MI	cTn elevation with ischemic symptoms or typical ECG changes, or by angiographic findings	MI: *n* = 134 Respiratory infectious disease: *n* = 13
**Violi et al., 2017, Clinical infectious diseases** [45]	CAP	MI during hospitalization for CAP	Third universal definition of AMI	MI: *n* = 89 (NSTEMI = 78 STEMI = 11) Respiratory infectious disease: *n* = 1182
**Musher et al., 2007, Clinical infectious diseases** [46]	Pneumococcal pneumonia	MI diagnosed at hospital admission for pneumonia	New ECG abnormalities (i.e., ST segment elevation or depression or Q waves) accompanied by cTn elevation	MI: *n* = 12 (NSTEMI = 9 STEMI = 3) Respiratory infectious disease: *n* = 170
**Corrales-Medina et al., 2015, Journal of American Medical Association** [47]	Pneumonia	MI and fatal coronary heart disease over 10 years after pneumonia hospitalization	Two algorithms based on symptoms, cardiac enzymes, and electrocardiographic evidence	MI: *n* = 247 Respiratory infectious disease: *n* = 1271
**Vejpongsa et al., 2019, The American Journal of Medicine** [48]	Acute influenza and other viral respiratory infections	Acute influenza and other viral infections in hospital admission for MI	ICD-9 diagnostic code	MI: *n* = 1,884,985 Respiratory infectious disease: *n* = 21,370 (Acute influenza = 9885 Other = 11,485)
**Caussin et al., 2015, International Journal of Cardiology** [49]	Wide spectrum of respiratory tract infection (flu-like illness with fever and sore throat), pneumonia or bronchitis	Possible exposure to respiratory infection within 35 days prior to admission for MI	Angiographically confirmed MI	MI: *n* = 578 Respiratory infectious disease: *n* = 123
**Ruane et al., 2017, Internal Medicine Journal,** [50]	Influenza	Association between STEMI and influenza epidemic	Angiographically confirmed STEMI (≤24 h) with at least ≥50% coronary stenosis	MI: *n* = 11,987 Respiratory infectious disease: *n* = NA (ERR 8.9)
**Peiris et al, 2003, The Lancet** [51]	SARS-CoV	Deaths for MI in hospitalized patients with SARS	Not reported	MI: *n* = 2 Respiratory infectious disease: *n* = 75
**Chong et al., 2004, Archives of Pathology and Laboratory Medicine** [52]	SARS-CoV	MI in post-mortem examinations for confirmed SARS infections	Post-mortem	MI: *n* = 2 Respiratory infectious disease: *n* = 8
**No data available**		MERS		NA

**Abbreviations:** CAP: community acquired pneumonia; CoV: coronavirus; cTn: cardiac troponin; GPRD: General Practice Research Database; ERR: excess relative risk; HMPV: human metapneumovirus; HPIV: human parainfluenza virus; ICD: international classification of diseases; MERS: Middle East respiratory syndrome; MI: myocardial infarction; NA: not available, NSTEMI: myocardial infarction without ST-segment elevation; RSV: respiratory syncytial virus; SARS-CoV: severe acute respiratory syndrome coronavirus; STEMI: ST-segment elevation myocardial infarction; UK: United Kingdom.

**Table 3 jcm-09-01683-t003:** Myocardial injury and ACS in patients with Covid-19—main studies.

Study, Year, Journal.	Population	Evaluation and Timeline	Cases with Myocardial Injury	Suspected ACS	in Hospital Mortality
**Huang et al., 2020, The Lancet** [5]	*n* = 41 ICU = 13 Non-ICU = 28	Myocardial injury = increased cardiac biomarkers or new ECG—echo abnormalities during hospitalization	*n* = 5 (12%) ICU = 4 (31%) Non-ICU = 1 (4%)	NA	*n* = 6 (15%)
**Wang et al., 2020, Journal of American Medical Association** [1]	*n* = 138 ICU = 36 Non-ICU = 102	Myocardial injury = increased cardiac biomarkers or new ECG—echo abnormalities during hospitalization	*n* = 10 (7.2%) ICU = 8 (22.2%) Non-ICU = 2 (2%)	NA	*n* = 6 (43%)
**Zhou et al., 2020, The Lancet** [66]	*n* = 191 Non-survivor = 54 Survivor = 137	Myocardial injury = increased cardiac biomarkers or new ECG—echo abnormalities during hospitalization	*n* = 33 (17%) Non-survivor = 32 (59%) Survivor = 1 (1%)	First autopsy performed = findings were consistent with AMI	*n* = 54 (28.3%)
**Shi et al., 2020, Journal of American Medical Association: Cardiology** [6]	*n* = 416	Myocardial injury = increased cardiac biomarkers regardless of new ECG—echo abnormalities during hospitalization	*n* = 82 (19.7%)	ECG features consistent with myocardial ischemia—NSTEMI: *n* = 14 (3.36%)	*n* = 57 (13.7%) With cardiac injury = 42 (51.2%) Without cardiac injury = 15 (4.5%)

**Abbreviations:** ACS: acute coronary syndromes; AMI: acute myocardial infarction; ECG: electrocardiogram; ICU: intensive care unit; NA: not available; NSTEMI: myocardial infarction without ST-segment elevation; STEMI: ST-segment elevation myocardial infarction.

**Table 4 jcm-09-01683-t004:** Drug treatment of acute coronary syndromes and coronary artery disease—evidence and potential interactions with drugs used to treat SARS-CoV-2 infection.

Therapy	Potential Interactions	Evidence	Notes
P2Y12 inhibitor: - Clopidogrel - Ticagrelor - Prasugrel	Lopinavir–Ritonavir	When coadministered with lopinavir–ritonavir, diminished effect of clopidogrel, increased effect of ticagrelor [101].	Consider using prasugrel if no contraindications [101]. Contraindications to prasugrel: previous intracranial hemorrhage, previous ischemic stroke or TIA, or ongoing bleeds; prasugrel is not recommended for patients >75 years of age or with a body weight <60 kg; or in NSTE-ACS if coronary anatomy is not known [112]. Contraindications for ticagrelor: previous intracranial hemorrhage or ongoing bleeds [112].
Aspirin	-	Lack of evidence on discontinuation of aspirin in Covid-19 patients.	Low-dose aspirin can be assumed to be safe as antiplatelet drug in Covid-19 patients [103].
Statins: - Atorvastatin - Rosuvastatin - Lovastatin - Simvastatin	Lopinavir–Ritonavir	When coadministered with lopinavir–ritonavir, increased effect of atorvastatin and rosuvastatin [101].	Start at lowest possible dose of rosuvastatin and atorvastatin and titrate, otherwise use pravastatin [101].
Beta-blockers: - Metoprolol - Carvedilol - Propranolol - Labetalol	Chloroquine–Hydroxychloroquine Fingolimod	Hydroxychloroquine has a potential role in reducing heart rate and may increase effect of beta-blockers [105].	When coadministered with chloroquine or hydroxychloroquine, beta-blockers dose reduction may be required [105].
ACEi/ARbs	-	No human evidence establishing a link between the use of these medications with an increased risk of Covid-19 acquisition or illness severity [108].	Abrupt withdrawal in high-risk patients, especially those who have heart failure or have had MI, may result in clinical instability and adverse outcomes [107,108].
Heparin	-	First evidences showed decreased mortality in severe Covid-19 patients with coagulopathy [111].	Given the interactions between some antiviral drugs and OACs, low molecular weight heparins, or unfractionated heparin should be preferred over OACs [101].

**Abbreviations**: ACEi: ACE inhibitors; ARBs: angiotensin receptor blockers; Covid-19: coronavirus disease; MI: myocardial infarction; NSAIDS: nonsteroidal anti-inflammatory drugs; NSTE-ACS: acute coronary syndromes without ST-segment elevation; OACs: oral anticoagulants; TIA: transient ischemic attack.

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
