# Peer review of "Acute Coronary Syndromes and Covid-19: Exploring the Uncertainties"

_jcm, 2020, doi:10.3390/jcm9061683_

Round 1

Reviewer 1 Report

The new SCAI and AHA guidelines recommend primary PCI over thrombolytics and this is relevant in USA. Some of the papers from China and Italy have recommended thrombolytics which has been cited by these authors but primary PCI remains the main form of recommended revascularization and that needs to be stressed in this review including recent citations to guidelines from SCAI. Once this is incorporated then we can consider accepting the mansucript

Reviewer 2 Report

This is an interesting review about Covid-19 and acute coronary syndromes.

This review is well documented and raises important questions. 

Comments:

- Table 1: "plaque rupture- érosion" appears twice

Reviewer 3 Report

Schievone  M and the coauthors discuss an important topic providing an overall review of a complex area.

In the title, the term “dark side of the outbreak”  is not very clear in its reference.  If the authors mean - "they are exploring the uncertainties", they should express it like that rather than using the term "dark side".

In Table 1:  "plaque rupture – erosion with occlusive thrombosis" is  mentioned under both "reduced myocardial perfusion" and "increased myocardial oxygen demand".  I presume it is a typo and should be removed from "increased oxygen demand"

The table detailing studies linking respiiratory infection with myocardial infarction is helpful however the authors should make clear how the diagnosis of myocardial infarction was made, especially since ECG changes are common with respiratory infection as are troponin elevations with sepsis. e.g. Lim W (Arch Intern Med  2006;166:2446) reported an average of 43% increased troponin with sepsis. 

The authors should reference 2 papers where coronary angiography was used to confirm coronary occlusion –  Ruane et al  Intern Med J 2017; 47: 522, and Caussin et al  Int J Cardiol 2015; 183:17 – and note whether any of the other papers also confirmed the ACS with coronary angiography.  If not, this should be listed as a limitation in view of the common troponin rise with sepsis.

The authors should also reference and discuss in more detail the apparent different relative risks of ACS with different organisms.  For instance Guan X  et al  showed a relative risk of 5.5 for Influenza A  but 20.4 for influenza B.   Inflamm Res 2012;61:591-598, and Kwong et al 2018 NEJM also showed a different relative risk of MI with different respiratory organisms.

In Table 4, the different column statements should all have references. The statement of evidence on aspirin is confusing.  The authors should use references regarding aspirin, and not quote “the media”.  The authors should also ideally avoid quoting Television statements (line 322) and instead reference a publication.

In their conclusion, it would be helpful if the authors diiscuss whether the evidence to date suggests that the troponin rise with COVID-19 is more an inflammatory response that a type 1 MI.  The recent NEJM article of angiographic findings in ST segment elevation with Covid-19 is also relevant (Bangalore et al  NEJM 2020-05-12.)  The authors mention a Peking guideline for use of thrombolytics, but should also discuss other guidelines considering angiography rather than thrombolysis, especially if if it uncertain whether there is coronary occlusion.

Reviewer 4 Report

Schiavone et al. present a narrative review describing the association between COVID-19 infection and acute cardiac events. The paper is ambitious in scope and a large number of very recently published papers have been cited. Unfortunately, due to the rapid pace of this pandemic most of this work is descriptive only with very little hard evidence to guide clinical decision making. This is reflected in the current manuscript which repeats these descriptions without any new insight provided. Consequently, it is difficult to identify what this manuscript has to add to prior work (e.g. https://doi.org/10.1161/CIRCULATIONAHA.120.047549). Some other issues warrant comment:

Major comments:

1) The title is misleading. Around the world, 100,000's of people have died from COVID-19-related disease. The majority of these deaths appear to arise from acute respiratory distress syndrome and respiratory failure. To suggest that acute coronary syndromes represent 'the dark side' of COVID-19 is absurd.

2) Although the title suggests the focus of the manuscript is acute coronary syndromes, much attention is (rightly) paid to acute myocardial injury which is not an acute coronary syndrome. Many of the publications cited make little effort to distinguish type 1 and type myocardial infarctions from myocardial injury and indeed often simply inaccurately classify ST elevation ACS and Non ST elevation ACS. This complexity and ambiguity has important clinical consequences as the evidence base for management of ACS (including routine invasive management) cannot be applied to non-ACS events associated with myocardial injury or infarction. For example citation 47 describes a paper where ICD-9 coding was used to identify ACS events. Such an approach cannot be applied to reliable differentiate these conditions.

3) The authors demonstrate a lack of understanding in this regard in reference to the same paper reporting a low rate of coronary catheterisation amongst patients with acute myocardial infarction in the context of viral influenza (22%). They then state that 'more than half' required revascularisation. In truth, amongst those patients who underwent coronary catheterisation the rate of revascularisation was lower in those with concomitant influenza than those without - suggesting that clinicians had appropriately identified which patients to put forward for coronary angiography.

Minor comments:

4) The term 'autoptic' is used throughout the manuscript. As a native English speaker this is an unusual word to use. I would suggest 'post-mortem' or 'autopsy' is used in place.

5) The grammar would benefit from thorough proof reading.

Round 2

Reviewer 4 Report

Thank you for your response to my comments and those of the other reviewers. The manuscript has been improved as a result.